# Safety and Immunogenicity of a Chimeric Subunit Vaccine against Shiga Toxin-Producing *Escherichia coli* in Pregnant Cows

**DOI:** 10.3390/ijms24032771

**Published:** 2023-02-01

**Authors:** Roberto M. Vidal, David A. Montero, Felipe Del Canto, Juan C. Salazar, Carolina Arellano, Alhejandra Alvarez, Nora L. Padola, Hernán Moscuzza, Analía Etcheverría, Daniel Fernández, Victoria Velez, Mauro García, Rocío Colello, Marcelo Sanz, Angel Oñate

**Affiliations:** 1Programa de Microbiología y Micología, Instituto de Ciencias Biomédicas, Facultad de Medicina, Universidad de Chile, Santiago 8380453, Chile; 2Instituto Milenio de Inmunología e Inmunoterapia, Facultad de Medicina, Universidad de Chile, Santiago 8380453, Chile; 3Programa de Inmunología, Instituto de Ciencias Biomédicas, Facultad de Medicina, Universidad de Chile, Santiago 8380453, Chile; 4Departamento de Microbiología, Facultad de Ciencias Biológicas, Universidad de Concepción, Concepción 4070386, Chile; 5Facultad de Ciencias Veterinarias, Universidad Nacional del Centro de la Provincia de Buenos Aires (UNCPBA), Tandil 7000, Argentina; 6Centro de Investigación Veterinaria de Tandil (CIVETAN), Universidad Nacional del Centro de la Provincia de Buenos Aires (UNCPBA), Tandil 7000, Argentina; 7Departamento de Clínica, Grupo de Medicina Veterinaria Traslacional (MEVET), Facultad de Ciencias Veterinarias, Universidad Nacional del Centro de la Provincia de Buenos Aires (UNCPBA), Tandil 7000, Argentina

**Keywords:** Shiga toxin-producing *Escherichia coli* (STEC), chimeric subunit vaccine, cattle immunization

## Abstract

Shiga toxin-producing *Escherichia coli* (STEC) is a zoonotic pathogen that causes gastroenteritis and Hemolytic Uremic Syndrome. Cattle are the main animal reservoir, excreting the bacteria in their feces and contaminating the environment. In addition, meat can be contaminated by releasing the intestinal content during slaughtering. Here, we evaluated the safety and immunogenicity of a vaccine candidate against STEC that was formulated with two chimeric proteins (Chi1 and Chi2), which contain epitopes of the OmpT, Cah and Hes proteins. Thirty pregnant cows in their third trimester of gestation were included and distributed into six groups (*n* = 5 per group): four groups were administered intramuscularly with three doses of the formulation containing 40 µg or 100 µg of each protein plus the Quil-A or Montanide™ Gel adjuvants, while two control groups were administered with placebos. No local or systemic adverse effects were observed during the study, and hematological parameters and values of blood biochemical indicators were similar among all groups. Furthermore, all vaccine formulations triggered systemic anti-Chi1/Chi2 IgG antibody levels that were significantly higher than the control groups. However, specific IgA levels were generally low and without significant differences among groups. Notably, anti-Chi1/Chi2 IgG antibody levels in the serum of newborn calves fed with colostrum from their immunized dams were significantly higher compared to newborn calves fed with colostrum from control cows, suggesting a passive immunization through colostrum. These results demonstrate that this vaccine is safe and immunogenic when applied to pregnant cows during the third trimester of gestation.

## 1. Introduction

Shiga toxin-producing *Escherichia coli* (STEC) is a zoonotic food-borne pathogen that causes diarrhea, dysentery and Hemolytic Uremic Syndrome (HUS), mainly in children under five years of age. STEC O157:H7 is by far the serotype most frequently implicated in severe disease and HUS worldwide. However, several non-O157 serotypes have emerged and are also an important cause of human disease in many countries [1].

The pathogenicity of STEC is mainly determined by the Shiga toxins (Stx), which cause inflammation in the intestinal mucosa. Once the Stx reach the bloodstream, they can cause tissue damage in the kidneys and central nervous system, as well as cause other unusual severe diseases complicated by multi-organ failure [2,3]. Furthermore, a number of different virulence factors are involved in the adherence of STEC to intestinal epithelial cells and colonization of the large intestine. For instance, the pathogenicity island (PAI) called the Locus of Enterocyte Effacement (LEE) encodes a type three secretion system (T3SS) that translocates effector proteins into the human enterocytes, leading to the lesion known as “attaching and effacing” (A/E). The A/E lesion is characterized by the loss of intestinal microvilli which leads to diarrhea [4]. The majority of the STEC strains associated with severe disease in humans harbor the LEE PAI. However, in the absence of the LEE, STEC strains can acquire and accumulate other PAIs, such as the Locus of Adhesion and Autoaggregation (LAA) [5], Subtilase-Encoding Pathogenicity Island (SE-PAI) [6] and Locus of Proteolysis Activity (LPA) [6], demonstrating the high genome plasticity and a wide variety of virulence factors that these pathogens possess. In fact, the LAA PAI promotes the intestinal colonization of STEC and has been identified among the LEE-negative strains associated with disease in humans [7].

STEC may be a resident or transient member of the gastrointestinal microbiota of several mammals, mainly livestock species. Of note, animals capable of maintaining STEC carriage without continuous exposure to the bacteria are defined as reservoirs [8]. Among them, cattle are considered the main reservoir and can intermittently shed STEC through their feces, contaminating the environment [9]. STEC strains have also been isolated from wild animals (e.g., rabbits, birds and rodents). However, it is not clear if they act as transient carriers or as reservoirs [10,11].

In general, intestinal STEC carriage in animals is asymptomatic because most of them lack vascular receptors for Stx; therefore, these toxins cannot be endocytosed and transported to extraintestinal tissues [12]. Occasionally, STEC can cause diarrhea in newborn calves. However, this pathology is thought to be associated with the A/E lesion and with an extensive bacterial colonization leading to the sloughing of enterocytes, rather than a direct cytotoxic action of the Stx [13,14]. Although cattle have receptors for Stx in intestinal epithelial cells, these toxins have been shown to have an immunosuppressive effect in these animals, apparently favoring intestinal colonization [15,16].

STEC carriage in cattle is determined by the ability of the bacteria to adhere to and colonize the large intestine. In particular, STEC O157:H7 have a tissue tropism for the recto-anal junction (RAJ), but non-O157:H7 serotypes may have tropism for other intestinal tissues [17,18]. In experimental infections of cattle and calves with STEC O157:H7, a number of LEE-encoded virulence factors, including Intimin and structural proteins of the T3SS, as well as non-LEE-encoded type III secreted effectors, have been shown to promote intestinal colonization [14,19,20]. However, the adherence of non-O157:H7 STEC to bovine intestinal epithelial cells appears to be mediated by mechanism distinct from those used by O157:H7 [21]. Thus, other proteins associated with adhesion phenotypes, such as EhaA, Iha, EspP and Efa-1, could also be important for the colonization of STEC in cattle [9]. Furthermore, there are STEC strains that persist in cattle for long periods of time, while other strains are identified only sporadically [22]. Consequently, the molecular mechanisms behind the ability of different STEC strains to persist in cattle and other reservoirs are not yet fully understood.

Cattle vaccination is a feasible intervention method to reduce the colonization and shedding of STEC and thus lower the risk of zoonotic transmission to humans [23]. Several vaccine candidates have been evaluated in controlled and natural conditions with variable results [24]. To date, two vaccines against STEC O157:H7 have been licensed for use in cattle in Canada and the USA; however, these formulations reduce the shedding but are not capable of completely clearing the colonization of these bacteria [24,25]. Besides, these vaccines have shown no efficacy in reducing the shedding of other STEC serotypes [26].

Previously, we developed a vaccine candidate against STEC based on two chimeric proteins (Chi1 and Chi2), which contain selected epitopes of the OmpT, Cah and Hes proteins. These proteins are ideal targets against STEC because they are immunogenic and widespread among LEE-positive and LEE-negative STEC strains isolated from humans [5,27,28]. Evaluation of this vaccine candidate in mice showed that it confers protection by reducing intestinal colonization of STEC O157:H7 and kidney damage caused by oral challenge with STEC O91:H21 [29]. Here, our objective was to evaluate the safety and immunogenicity of this vaccine candidate in pregnant cows. Overall, our results demonstrate the safety and immunogenicity of this STEC vaccine. These data need to be complemented by effectiveness and challenge studies to formulate a recommendation protocol for use in cattle.

## 2. Results

### 2.1. Experimental Design

The study presented here is a double-blind, randomized field trial, the objectives of which were to evaluate the safety and immunogenicity of a vaccine candidate against STEC in pregnant cows. This vaccine was previously shown to be protective in a murine model [29]. Figure 1 shows the experimental design of this study.

We implemented a dose-escalation scheme to determine the optimal safe and immunogenic dose of this vaccine candidate. For this, four different formulations were generated containing two increasing concentrations (40 µg and 100 µg) of the Chi1 and Chi2 proteins and the Quil-A or Montanide™ Gel adjuvants (Figure 1). The placebos contained PBS plus the corresponding adjuvant.

A total of 30 pregnant cows were randomly assigned to six groups (*n* = 5 per groups) to receive the vaccine formulations (P1, P2, P4 and P5 groups) or placebos (P3 and P6 control groups). The immunization protocol was a three-dose schedule (2 weeks apart) by intramuscular route.

### 2.2. Safety

All the animals showed good health conditions throughout the field trial. No clinically relevant differences were documented with respect to local and systemic adverse events between animals that received the highest vaccine dose (P2 and P5 groups) and animals that received the standard dose (P1 and P4 groups) or placebos (P3 and P6 groups).

Additionally, no significant differences were observed in rectal temperatures and weights after administration of the different treatments (Appendix A). The hematological and blood biochemical parameters evaluated remained within the reference range and no significant differences were observed between groups at any time (Appendix A).

The calves were born at term, healthy and with normal weight, except for one calf from the P1 group that died during parturition due to fetal dystocia and one from the P3 group that was aborted (Appendix A). Thus, these results indicate that all vaccine formulations were safe and well tolerated.

### 2.3. Serum Antibody Response

Before the first immunization, fecal samples of all animals were analyzed by PCR for the presence of STEC, and the herd was found to be positive (result not shown). Consistent with the above, pre-immune sera from all animals were seropositive to STEC, having a baseline of Chi1/Chi2-specific IgG and IgA antibodies. However, the levels of these antibodies were similar among the groups (Figure 2).

After three immunizations with the vaccine formulations, a significant increase in serum IgG levels against both proteins (Figure 2a,b), but not the IgA antibodies (Figure 2c,d), was observed compared to the control groups. The increased level of specific IgG in sera was similar between animals receiving the standard or the highest dose of the formulation, regardless of the adjuvant used. In fact, no significant differences were observed in groups receiving vaccine formulations containing the Quil-A adjuvant (P1 and P2) compared to groups receiving vaccine formulations containing MontanideTM Gel adjuvant (P4 and P5). Therefore, all vaccine formulations triggered specific IgG antibodies in sera at similar levels.

### 2.4. Colostrum Antibody Response

Bovine colostrum may contain antibodies against important virulence factors of STEC [30]. Therefore, we determined the levels of Chi1/Chi2-specific antibodies in the colostrum of immunized cows. As a result, we found a slight but significant difference in the specific IgG levels between the P2 group and the P3 control group (Figure 3a). Additionally, although there were no significant differences, a trend towards higher specific IgG levels was observed in the P4 and P5 groups compared to the P6 control group (Figure 3b). By contrast, specific IgA antibody levels in colostrum were variable, and no significant differences were observed among the groups.

It has been reported that STEC-specific colostral antibodies are transferred to newborn calves after feeding them with colostrum [31,32]. Thus, passive immunization of newborn calves could reduce STEC colonization and therefore constitutes an interesting strategy for the control of this pathogen.

Notably, our results showed that calves fed with colostrum from their immunized dams had significantly higher levels of anti-Chi1/Chi2 IgG antibodies in serum compared with calves fed with colostrum from control cows (Figure 4). This was especially evident in the P4 and P5 groups compared to the P6 control group. Regarding the anti-Chi1/Chi2 IgA antibodies in serum, a significantly higher level was found in the newborn calves in the P4 group compared to the P6 control group (Figure 4b). In the other groups, there were no differences in the levels of specific IgA in serum.

## 3. Discussion

Most of the vaccine candidates against STEC that have been evaluated in cattle are based on virulence factors encoded on the LEE PAI [24,25,32,33,34]. By contrast, the vaccine candidate evaluated here contains epitopes of OmpT, Cah and Hes proteins. To our knowledge, no other vaccine candidate has evaluated the use of these antigens.

These three proteins are suitable targets to develop preventive therapies against STEC infections because: (i) their encoding genes are conserved, widely distributed and highly frequent in both LEE-positive (OmpT, Cah) and LEE-negative STEC strains (Hes, Cah and OmpT) [27]; (ii) they are expressed in vivo during infection in humans and are reactive to hyperimmune sera from HUS patients; and (iii) they have been shown to participate in different colonization-associated phenotypes in vitro assays and in infections of human epithelial cells [5,35,36]. However, it is important to note that the role, if any, of these proteins in the colonization of cattle is currently unknown.

Previously, this vaccine candidate was evaluated in a murine model of STEC infection, with promising results [29]. In the present study, we tested the safety and immunogenicity of this vaccine in pregnant cows. For this, the vaccine was formulated with two different concentrations, a standard dose of 80 µg and a higher dose of 200 µg of the Chi1 and Chi2 proteins. We used the Quil-A and the Montanide™ Gel adjuvants because they are safe and have adjuvant activity in cattle [37,38].

Our results indicate that all four vaccine formulations were well tolerated and did not cause adverse reactions, fetal malformations or abortions. The abortion observed during this study occurred in the P3 control group, and its cause was not determined; therefore, it was not associated with the vaccine administration. In this respect, pregnancy loss rate varies between 15% and 23% in livestock and may be due to a variety of causes and combination of factors [39,40,41,42]. This result demonstrates the safety of the vaccine formulations when administered to pregnant cows in their third trimester and following the described immunization protocol.

Based on PCR detection, we found that the cows included in this study were colonized by STEC before the first immunization (not shown). Cattle in Buenos Aires province have a prevalence of STEC as high as 63% [43]; therefore, cattle under natural conditions in this region are highly likely to be colonized with STEC. Thus, it is expected that cows in our study have been exposed to a plethora of diverse STEC strains during their lifetime.

In line with this, we found pre-existing anti-Chi1/Chi2 antibodies in the preimmune sera from the cows (Figure 2). This suggests that, as occurs in humans, the OmpT, Cah and Hes proteins are also expressed by STEC during cattle infection. Many studies have also found a baseline of STEC-specific antibodies in cattle naturally infected with STEC [24,44]; however, despite the pre-existence of these antibodies, cattle are susceptible to being colonized by STEC, often resulting in persistent shedding of these bacteria. Thus, circulating STEC-specific antibody levels should be analyzed with caution, as they should not be considered an absolute correlate of immunity.

Therefore, for the optimal evaluation of the immunogenicity of a vaccine against STEC, both the determination of systemic antibodies and secreted antibodies at the site of colonization must be considered, together with cellular immune responses [16]. Nevertheless, the objective of this work was to carry out an initial evaluation of the immunogenicity and optimal dose of our vaccine candidate.

In this sense, there are some results regarding the immunogenicity of our vaccine that deserve to be highlighted. First, vaccine formulations containing 80 µg of chimeric proteins elicited significant levels of specific IgG antibodies that were similar to those obtained with formulations containing 200 µg (Figure 2a,b). This would favor the scaling and costs of production of this vaccine.

Second, a trend for a higher level of anti-Chi1/Chi2 IgG antibodies in colostrum was observed in the immunized groups compared to the control groups (Figure 3). This difference was statistically significant only between the P2 group and the P3 control group. However, there was a high variability in the levels of antibodies in colostrum, which possibly affected the power of the statistical test used. In particular, we consider that this variability could be attributed to the technical difficulty of taking colostrum samples from cows kept under natural conditions.

Third, although the humoral responses in colostrum were inconclusive, the levels of anti-Chi1/Chi2 IgG antibodies in the sera of newborn calves from immunized cows were significantly higher than those found in sera of newborn calves from control cows (Figure 4). Remarkably, significant levels of specific IgA were also observed in the serum of calves from the P4 group compared to the P6 control group. In humans and other primates, the transfer of IgG from the mother to the neonate occurs prenatally (across the placenta) and postnatally (through lactation). In ruminants, the placenta transmits little or no IgG antibodies [45,46]. Instead, there is exhaustive evidence showing that the mammary gland of these species secretes large amounts of IgG during colostrum formation [45]. Consistent with these studies, it has been shown that STEC-specific antibodies in colostrum from naturally infected or vaccinated cows are efficiently transferred to the newborn calves by feeding with hyperimmune colostrum [31,32,34,47]. Thus, the higher levels of anti-Chi1/Chi2 IgG antibodies in the serum of newborn calves from immunized cows strongly suggest that a passive immunization through colostrum occurred. Since cattle during the first months of life are rapidly colonized by STEC [48], passive immunization of calves may be an alternative strategy to prevent the early colonization by these bacteria [32].

In conclusion, taken together, the results of this study demonstrate the safety of this vaccine candidate against STEC and provide information on its immunogenicity when administered to pregnant cows in their third trimester of gestation.

## 4. Materials and Methods

### 4.1. Animals

All animal experiments and protocols were approved (Approval Code: Acta de Bienestar Animal ResCA 087/02) by the Animal Welfare Commission at the National University of the Center of the Province of Buenos Aires, Tandil, Argentina. A total of 30 Aberdeen Angus pregnant cows (third trimester of pregnancy) were obtained, housed and fed with standard food and drink on a cattle farm located in Tandil, Argentina. Before the first immunization, fecal samples were taken and analyzed by PCR (using protocol described in [49]) for the presence of STEC, and the herd was found to be positive. The general characteristics of the cows are shown in Appendix A.

### 4.2. Production of Vaccine Formulations and Immunization Protocol

Chimera 1 (Chi1) and Chimera 2 (Chi2) proteins were produced and purified as described previously [29]. The vaccine formulations and placebos were sterilized by filtration and packaged at a veterinary pharmaceutical laboratory under good manufacturing practices. The vaccine formulations contained 80 (40 µg Chi1 + 40 µg Chi2) or 200 µg (100 µg Chi1 + 100 µg Chi2) of chimeric proteins plus the Quil-A or Montanide™ Gel adjuvants, while placebos contained phosphate-buffered saline (PBS) solution plus the corresponding adjuvant. The animals were ear-tagged identified and randomly assigned to receive a vaccine formulation or placebo in a three-dose schedule (2 weeks apart), which were administered intramuscularly in the cervical region. Four groups (P1, P2, P4 and P5; *n* = 5 per group) received the vaccine formulations and two control groups (P3 and P6; *n* = 5 per group) received adjuvants only (Figure 1, Appendix A).

Basic clinical observation and general adverse reactions were evaluated by measuring rectal temperature, weight and loss of appetite. Local adverse reactions at the injection site such as swellings or pain were evaluated by palpation. Local inflammation was examined by palpation of the prescapular lymph node.

### 4.3. Sample Collection

Blood samples (13 mL) were collected by venipuncture into EDTA tubes before each immunization and 2 weeks after the last boost. These samples were used to determine hematological and blood biochemical parameters. To obtain sera, blood samples (7 mL) were collected into non-anticoagulant tubes and then left at 37 °C for 30 min and centrifuged at 1000× *g* for 10 min, and the supernatants (sera) were collected and stored at −20 °C until use. Additionally, blood samples were taken from the newborn calves during the first 3 days of life and subsequently processed to obtain sera. Colostrum samples were obtained during the prodromal state of labor or within the first 24 h after parturition. Colostrum samples were delipidated by centrifugation (1000× *g* at 4 °C for 45 min), and the watery phase was stored at −20 °C until use.

### 4.4. Hematological and Blood Biochemical Parameters

Hematological and blood biochemical parameters were determined by using the Hemax330 hematology analyzer (B&E Scientific Instrument Co., Ltd., Yantai, China) and the CM 250 Wiener Lab^®^ automatic clinical chemistry analyzer (Wiener Lab Group, Santa Fe, Argentina), respectively. Hematological parameters analyzed were red blood cell count, hematocrit, hemoglobin, mean corpuscular volume, mean corpuscular hemoglobin (MCH) mean corpuscular hemoglobin concentration (MCHC), white blood cell count, immature neutrophils, segmented neutrophils, lymphocytes, monocytes, eosinophils and basophils. Biochemical parameters evaluated were total proteins, albumins, globulins, albumin/globulin ratio, alkaline phosphatase, creatinine, alanine aminotransferase and blood urea nitrogen.

### 4.5. Humoral Immune Responses

The humoral immune responses to the vaccine formulations were determined by indirect ELISA. For this, 96-well ELISA plates (Nunc-Immuno Plates, ThermoFisher, Waltham, MA, USA) were incubated with a mixture of 1 µg of Chi1 and Chi2 proteins diluted in 100 µL of phosphate saline buffer (PBS; 1X, pH 7.2) overnight at 4 °C. Plates were washed 3 times with PBS (400 µL/well) containing 0.05% Tween 20 (T-PBS). Plates were then incubated with 300 µL/well of blocking solution (T-PBS + 0.5% bovine serum albumin) for 15 min at room temperature. Animal sera were diluted 1:50 to 1:1600 (for IgA measurement) or 1:250 to 1:16,000 (for IgG measurement) in blocking solution (100 µL/well) and incubated for 60 min at 37 °C. Colostrum samples were diluted 1:200 (for IgA measurement) or 1:1000 (for IgG measurement) and incubated as described above. After 5 washes with T-PBS (400 µL/well), anti-Bovine IgA (Fc)-HRP (NB773, Novus Biologicals, Centennial, CO, USA) or anti-Bovine IgG (Fc)-HRP (SAB3700020, Sigma-Aldrich, St. Louis, MO, USA), diluted 1:1000 in blocking solution (100 µL/well), were incubated for 60 min at 37 °C. After 5 washes with T-PBS (400 µL/well), plates were incubated with the 3,3′,5,5′-tetramethylbenzidine (TMB) liquid substrate (T0440, Sigma-Aldrich, USA) for 10 min at room temperature. Absorbance was determined at 405 nm using a Synergy HT microplate reader (Biotek Instruments, Winooski, VT, USA). Each sample was determined in duplicate and with at least three independent replicates.

### 4.6. Statistical Analysis

Statistical differences in anti-Chi1/Chi2 IgG sera and IgA antibody were analyzed by a two-way ANOVA, followed by Tukey’s multiple comparison test. Statistical differences in colostrum anti-Chi1/Chi2 IgG and IgA antibodies were analyzed by the Kruskal−Wallis test, followed by Dunn’s multiple comparison test. For all statistical tests, a *p* value of <0.05 was considered significant.

## Figures and Tables

**Figure 1 ijms-24-02771-f001:**
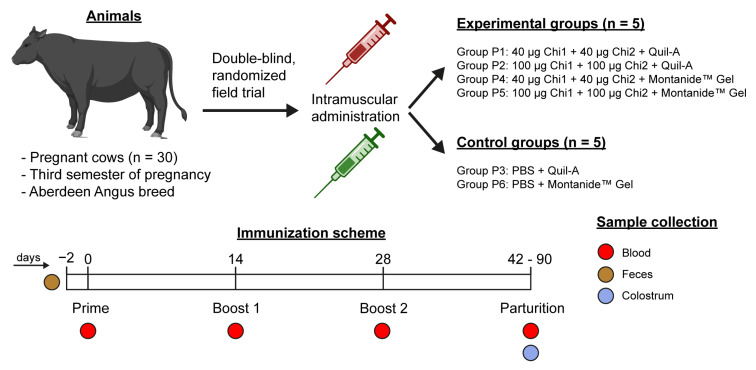
Experimental design of this field trial. The different groups of animals, vaccine formulations and placebos and immunization schedule are shown.

**Figure 2 ijms-24-02771-f002:**
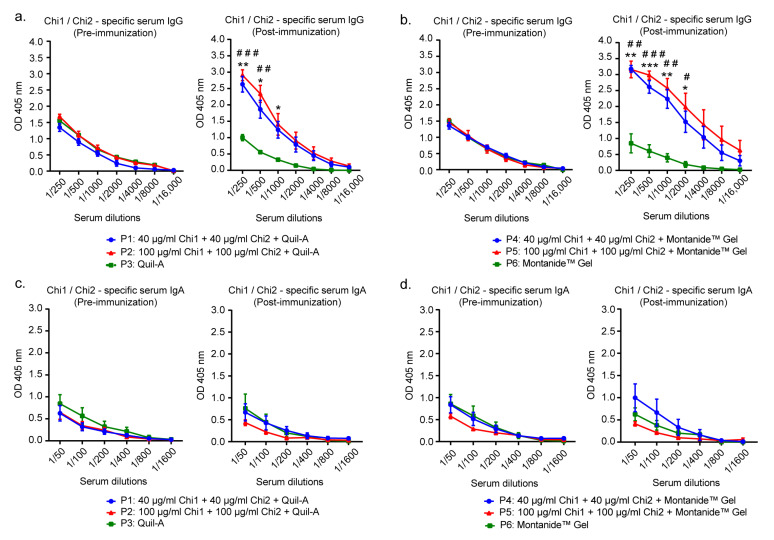
Serum antibody response in cows immunized with the vaccine formulations. Sera obtained before the first immunization and after the last immunization were used for the determination of anti-Chi1/Chi2 IgG and IgA antibodies. Samples were analyzed in duplicate, and the results are expressed as means ± SEM of absorbance values at 405 nm for each serum dilution, *n* = 5 animals per group. Anti-Chi1/Chi2 IgG (**a**) and IgA (**c**) antibodies from P1, P2 and P3 groups. Anti-Chi1/Chi2 IgG (**b**) and IgA (**d**) antibodies from P4, P5 and P6 groups. Statistical analysis was performed using a two-way ANOVA, followed by Tukey’s multiple comparison test. *p* < 0.05 was considered significant. Asterisks (*) indicate significant differences between the group immunized with the formulation containing 40 µg of each chimeric protein and the control group. * *p* < 0.05, ** *p* < 0.005, *** *p* < 0.0005. Number signs (#) indicate significant differences between the group immunized with the formulation containing 100 µg of each chimeric protein and the control group. # *p* < 0.05, ## *p* < 0.005, ### *p* < 0.0005.

**Figure 3 ijms-24-02771-f003:**
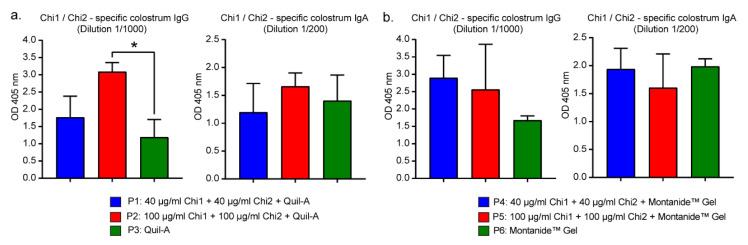
Chi1/Chi2—specific antibody levels in colostrum. (**a**) Colostrum from P1, P2 and P3 groups. (**b**) Colostrum from P4, P5 and P6 groups. Data analysis was by Kruskal−Wallis test, followed by Dunn’s multiple comparison test. *p* < 0.05 was considered significant. * *p* < 0.05.

**Figure 4 ijms-24-02771-f004:**
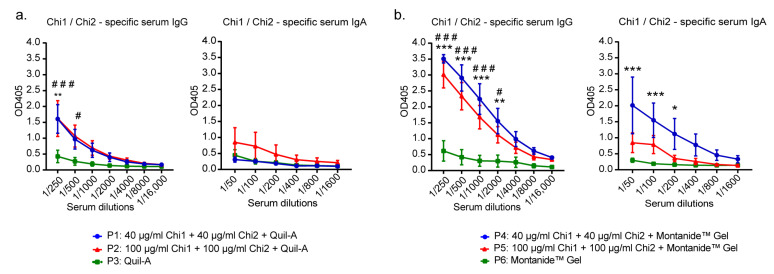
Chi1/Chi2—specific antibody levels present in the serum of newborn calves. (**a**) Newborn calves from groups P1, P2 and P3. (**b**) Newborn calves from groups P4, P5 and P6. Data analysis was by a two-way ANOVA, followed by Tukey’s multiple comparison test. *p* < 0.05 was considered significant. Asterisks (*) indicate significant differences between the group immunized with the formulation containing 40 µg of each chimeric antigen and the control group (adjuvant alone). * *p* < 0.05, ** *p* < 0.005, *** *p* < 0.0005. Number signs (#) indicate significant differences between the group immunized with the formulation containing 100 µg of each chimeric antigen and the control group (adjuvant alone). # *p* < 0.05, ### *p* < 0.0005.

## Data Availability

The data that support the findings of this study are available on request from the corresponding author R.V. Amino acid sequences of Chimeric proteins and identified epitopes are not publicly available due to legal restrictions and an ongoing international patent application.

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
