# Peer review of "Safety and Immunogenicity of a Chimeric Subunit Vaccine against Shiga Toxin-Producing *Escherichia coli* in Pregnant Cows"

_ijms, 2023, doi:10.3390/ijms24032771_

Round 1

Reviewer 1 Report

The article is well written and addresses an important topic for veterinary medicine. The text reads with pleasure and I think many readers will also welcome it. Some detail is required:

-        in accordance with accepted standards, it will be easier to use the resulting data in commercialization if it is clear that it was conducted in accordance with OECD standards. Please indicate the relevant OECD schemes;

-        please provide ethics committee approval number;

-        the aim of the study and the research hypothesis should be clearly formulated;

-        it may be worth noting in the discussion the possibility of infection of wild animals (e.g. rodents) and transmission of the pathogen;

-        in terms of the layout of the text, I suggest that only the data obtained in the survey be included in the results section, and the discussion elements be moved to the "discussion" section e.g. “Pregnancy loss rate varies between 15% and 23% in livestock and may be due to a variety of causes and combination of factors [28–31] ” or “This was an expected result since cattle in Buenos Aires province have a high prevalence of STEC colonization [32]”.

Author Response

Responses to reviewers of Vidal RM et al.: “Safety and immunogenicity of a chimeric subunit vaccine against Shiga toxin-producing Escherichia coli in pregnant cows”.

Manuscript ID: ijms-2123718

We want to thank the reviewers for their insightful comments. The authors fully agreed with their comments and felt that their suggestions and recommendations greatly increased the quality of the manuscript. Our specific responses to the reviewers’ comments are listed below.

Response to Reviewer 1 comments

The article is well-written and addresses an important topic for veterinary medicine. The text reads with pleasure, and I think many readers will also welcome it. Some detail is required:

Point 1: in accordance with accepted standards, it will be easier to use the resulting data in commercialization if it is clear that it was conducted in accordance with OECD standards. Please indicate the relevant OECD schemes.

Response 1: We are grateful for the suggestion on the OECD schemes and the projection for the commercialization of our vaccine. However, we are focused on doing more studies on our vaccine and its protection against STEC colonization. We hope to have the necessary experimental results to work on the vaccine's licensing and approval process in the medium term.

Point 2: please provide ethics committee approval number.

Response 2: All animal experiments and protocols were approved (Approval Code: Acta de Bienestar Animal ResCA 087/02) by the Animal Welfare Commission at the National University of the Center of the Province of Buenos Aires, Tandil, Argentina. This information was included in lines 301 – 302. In addition, upon request, the document was sent to the IJMS Assistant Editor.

Point 3: the aim of the study and the research hypothesis should be clearly formulated.

Response 3: The objective was indicated in lines 109 - 110. The hypothesis was not indicated because this is a descriptive study.

Point 4: it may be worth noting in the discussion the possibility of infection of wild animals (e.g. rodents) and transmission of the pathogen.

Response 4: This information was added in lines 71 – 73.

Point 5: in terms of the layout of the text, I suggest that only the data obtained in the survey be included in the results section, and the discussion elements be moved to the "discussion" section e.g. “Pregnancy loss rate varies between 15% and 23% in livestock and may be due to a variety of causes and combination of factors [28–31] ” or “This was an expected result since cattle in Buenos Aires province have a high prevalence of STEC colonization [32]”.

Response 5: The indicated information was moved to the discussion. (Lines 242 – 246; lines 249-252)

Reviewer 2 Report

This study aimed to determine the Safety and immunogenicity of a chimeric subunit vaccine 2 against Shiga toxin-producing Escherichia coli in pregnant 3 cows.

General comments

The manuscript is well written and well organized.

Minor comments.

Line 51-54: The sentence is too ling please shorten

Line 76-79:  The sentence is too ling please shorten

Author Response

Responses to reviewers of Vidal RM et al.: “Safety and immunogenicity of a chimeric subunit vaccine against Shiga toxin-producing Escherichia coli in pregnant cows”.

Manuscript ID: ijms-2123718

We want to thank the reviewers for their insightful comments. The authors fully agreed with their comments and felt that their suggestions and recommendations greatly increased the quality of the manuscript. Our specific responses to the reviewers’ comments are listed below.

Response to Reviewer 2 comments

This study aimed to determine the Safety and immunogenicity of a chimeric subunit vaccine against Shiga toxin-producing Escherichia coli in pregnant cows.

General comments

The manuscript is well written and well organized.

Minor comments.

Point 1: Lines 51-54: The sentence is too long, please shorten

Response 1: The sentence was shortened.

Point 2: Lines 76-79:  The sentence is too long, please shorten

Response 2: The sentence was shortened.

Reviewer 3 Report

In general the manuscript is correctly written. The current manuscript is interesting and novel, so due to that the manuscript could of valuable contribution to the scientific community. Generally, the study appear to be sound, well-designed and could be also of practical value. The use of English is mostly very good. Therefore, the undertaken issue is interesting and worth of disseminating

Author Response

Responses to reviewers of Vidal RM et al.: “Safety and immunogenicity of a chimeric subunit vaccine against Shiga toxin-producing Escherichia coli in pregnant cows”.

Manuscript ID: ijms-2123718

We want to thank the reviewers for their insightful comments. The authors fully agreed with their comments and felt that their suggestions and recommendations greatly increased the quality of the manuscript. Our specific responses to the reviewers’ comments are listed below.

Response to Reviewer 3 comments

In general, the manuscript is correctly written. The current manuscript is interesting and novel, so due to that the manuscript could of valuable contribution to the scientific community. Generally, the study appear to be sound, well-designed and could be also of practical value. The use of English is mostly very good. Therefore, the undertaken issue is interesting and worth of disseminating. 

Response: We appreciate the reviewer's kind comments and agree that our work will contribute to the scientific community interested in STEC control and prevent human infection.
